# Dual Coating of Chitosan and Albumin Negates the Protein Corona-Induced Reduced Vascular Adhesion of Targeted PLGA Microparticles in Human Blood

**DOI:** 10.3390/pharmaceutics14051018

**Published:** 2022-05-09

**Authors:** Genesis Lopez-Cazares, Omolola Eniola-Adefeso

**Affiliations:** 1Department of Chemical Engineering, University of Michigan, Ann Arbor, MI 48109, USA; glope@umich.edu; 2Biointerfaces Institute, University of Michigan, Ann Arbor, MI 48109, USA

**Keywords:** PLGA, chitosan, human serum albumin, particle adhesion, protein corona

## Abstract

Vascular-targeted carriers (VTCs) have the potential to localize therapeutics and imaging agents to inflamed, diseased sites. Poly (lactic-co-glycolic acid) (PLGA) is a negatively charged copolymer commonly used to construct VTCs due to its biodegradability and FDA approval. Unfortunately, PLGA VTCs experienced reduced adhesion to inflamed endothelium in the presence of human plasma proteins. In this study, PLGA microparticles were coated with chitosan (CS), human serum albumin (HSA), or both (HSA-CS) to improve adhesion. The binding of sialyl Lewis A (a ligand for E-selectin)-targeted PLGA, HSA-PLGA, CSPLGA, and HSA-CSPLGA to activated endothelial cells was evaluated in red blood cells in buffer or plasma flow conditions. PLGA VTCs with HSA-only coating showed improvement and experienced 35–52% adhesion in plasma compared to plasma-free buffer conditions across all shear rates. PLGA VTCs with dual coating—CS and HSA—maintained 80% of their adhesion after exposure to plasma at low and intermediate shears and ≈50% at high shear. Notably, the protein corona characterization showed increases at the 75 and 150 kDa band intensities for HSA-PLGA and HSA-CSPLGA, which could correlate to histidine-rich glycoprotein and immunoglobulin G. The changes in protein corona on HSA-coated particles seem to positively influence particle binding, emphasizing the importance of understanding plasma protein–particle interactions.

## 1. Introduction

Vascular-targeted carriers (VTCs), designed for intravenous (IV) administration, have the potential to effectively transport therapeutics and imaging agents to disease sites that typically require invasive surgery, including hard-to-reach tumors and cardiovascular disease (CVD) [1]. By virtue of their surface ligands, VTCs can enhance tissue specificity and local drug concentration via disease biomarkers present on the vasculature and surrounding target tissues, leading to reduced off-target toxicity. Particulate carriers (e.g., micelles, liposomes, and polymeric particles) are of great interest for VTCs due to their ability to encapsulate drugs, protecting them from degradation. Micelles and liposomes are highly biocompatible carriers constructed from lipids and amphiphilic molecules but are restricted to encapsulating only hydrophobic or hydrophilic drugs, respectively [2]. Liposomes are the most widely studied VTCs and have been successfully translated into the clinic mainly for the treatment of cancer [3]. However, their instability in physiological mediums, lack of controlled release, and fast oxidation of some phospholipids that can affect their storage are of current concern [4,5].

Polymeric particles are a potential alternative due to their tunable properties that demonstrate improved drug stability, effective controlled release, and enhanced resistance to degradation over other carrier types [6,7,8]. Synthetic polymers are primarily employed to construct injectable pharmaceutical products due to their biodegradability, biocompatibility, and FDA approval for numerous other clinical applications [9]. Poly (lactic-co-glycolic acid) (PLGA)—a negatively charged polymer that is easily degraded via hydrolysis into its monomeric forms that are subsequently metabolized by the human body [10]—is the most widely studied material because of its tunable capability to encapsulate various drugs, ranging from small molecules to proteins [11,12]. Despite the benefits of PLGA, only 19 injectable long-acting formulations have been authorized for clinical use since its FDA approval in 1989, and most are designed for intramuscular administration [13,14].

One main obstacle facing the clinical translation of intravenously administered PLGA carriers is the insufficient understanding of interactions between PLGA and biological environments in systemic circulation. Notably, once injected, a VTC must navigate the complex blood flow environment, marginating (or localizing) from the red-blood-cell-rich core to the vascular wall for interaction with the targeted biomolecules presented by the endothelium. In this regard, many parameters can affect the functionality of VTCs, including their size, shape, and material properties. For instance, microparticles in the 2–3 µm range marginate more efficiently than their nanoparticle counterparts, which become trapped in the red-blood-cell-rich core [15,16]. Recently, researchers have demonstrated that high protein adsorption onto PLGA VTC surface, specifically immunoglobulins A and M, drastically reduces particle binding to the endothelium in human whole blood or plasma conditions [17,18,19,20]. The addition of polyethylene glycol (PEG) onto the surface of drug carriers is commonly used to reduce protein adsorption, leading to their increased systemic circulation [21]. Unfortunately, PEGylation of the PLGA surface did not improve its adhesion to inflamed endothelial cells in the presence of plasma proteins [17,19]. Moreover, PEG-coated polymeric particles exhibited increased uptake by neutrophils in human blood, which could prevent carriers from reaching diseased target sites [22]. Thus, in this work, alternate surface coatings for PLGA VTCs were explored for improved vascular wall adhesive interactions under human blood flow conditions—specifically, chitosan (CS) and human serum albumin (HSA).

Natural biodegradable polymers, such as CS and HSA, have been used to develop drug carriers due to their abundance, biocompatibility, and low toxicity [23]. The attachment of CS and HSA to the surface of drug carriers has been employed to improve biological interactions serving as potential alternatives to PEG [24,25]. Chitosan is a positively charged polysaccharide with high solubility in acidic conditions and can be degraded by multiple enzymes [26,27]. Alternatively, HSA is the most abundant protein in the blood and has dysopsonin properties [28]. The addition of CS or HSA onto the surface of particle carriers has demonstrated many advantages, including improved physicochemical stability, extended circulation time similar to PEGylation, and controlled drug release for a wide range of drug delivery applications [23,24,25,29]. Here, PLGA particles were coated with CS, HSA, or both (HSA-CS) to mitigate PLGA’s low adhesivity and high protein adsorption in human blood. CS was physically adsorbed during particle fabrication, and HSA was covalently attached to the surface of PLGA or CSPLGA. Then, sialyl Lewis A (sLe^A^) or human anti-ICAM1 (aICAM1) was coupled onto the particles to target E-selectin or ICAM-1 expressed by endothelial cells during inflammation. The adhesion of targeted PLGA, HSA-PLGA, CSPLGA, and HSA-CSPLGA to an inflamed endothelial cell monolayer was evaluated in human plasma. A significant improvement in particle binding for sLe^A^-targeted HSA-PLGA and HSA-CSPLGA over PLGA was seen, which may be linked to the observed differences in the protein corona composition, particularly at the 75 and 150 kDa molecular weights. Ultimately, the present results will contribute information on the utility of CS and HSA as coatings for PLGA microparticles for vascular-targeted drug delivery.

## 2. Materials and Methods

### 2.1. Fabrication of Uncoated and CS-Coated PLGA Microparticles

PLGA microparticles were fabricated using the emulsion solvent evaporation method established in the literature [12,30]. First, 100 mg of low molecular weight PLGA (50:50, 0.15–0.25 dL/g, ≈6.4 kDa; Lactel Absorbable Polymers, Birmingham, AL, USA) was dissolved in 18 mL of methylene chloride (Fisher Chemical, Pittsburgh, PA, USA) to form the oil phase. The water phase for fabricating uncoated PLGA particles contained 1.25 g of polyvinyl alcohol (PVA) (30–70 kDa; Millipore Sigma, Burlington, MA, USA) dissolved in 250 mL of deionized water at 150 °C for 2 h. The solution was allowed to cool, diluted to 250 mL for a 0.5% *w*/*v* PVA concentration, and filtered before use. For particle fabrication, 75 mL of the PVA water phase was placed in a 400 mL beaker and continuously stirred at high speed on an overhead mixer with a glass propeller. The oil phase was slowly injected with a glass syringe into the water phase, and the emulsion was continuously stirred for 2 h to allow the solvent to evaporate. Particles of approximately 2 µm mean diameter were obtained via centrifugation wash steps and freeze-dried overnight in a Labconco lyophilizer. Dried particles were stored at −20 °C until use. A low amount of rhodamine was added to the oil phase with the PLGA polymer to obtain fluorescent particles. The same emulsion solvent evaporation method for uncoated PLGA was followed to fabricate CS-coated PLGA (CSPLGA), except the water phase was prepared by dissolving 1 g of low molecular weight CS (5–20 mP·s, TCI America, Portland, OR, USA) in 250 mL of 0.3 M HCl solution (pH ≈ 1) at 150 °C. The CS solution was allowed to cool, diluted to a 0.4% *w*/*v* CS concentration, had its pH adjusted to 4.5 with 10 M NaOH, and filtered before use. The oil phase preparation and particle fabrication steps remained the same as described for uncoated PLGA.

### 2.2. HSA Conjugation and Staining

HSA attachment onto PLGA and CSPLGA particles was carried out using carbodiimide chemistry to obtain HSA-PLGA and HSA-CSPLGA, respectively. Briefly, HSA was reacted to the carboxylic acids on PLGA or amines on CSPLGA in one reaction step, enabled by the fact that proteins have an abundance of both amine and carboxylic acid groups at the C-terminus, N-terminus, and amino acid side chains [31]. Dried particles were resuspended in deionized water and sonicated at 20% amplitude to disperse, and concentration was obtained using a hemocytometer. HSA from human plasma (Millipore Sigma, ≥95%) was dissolved in filtered deionized water at 20 mg/mL. Next, 5 × 10^6^ particles were added to 0.4 mL of HSA solution, diluted to 0.5 mL, and placed on a rotator for 20 min to allow HSA to adsorb onto the particle surface. Then, N-(3-dimethylaminopropyl)-N′-ethyl carbodiimide hydrochloride (EDAC, Millipore Sigma) was dissolved in a 200 mM MES buffer at pH 7.2 at 100 mg/mL. Next, 0.5 mL of EDAC/MES solution was added to particles incubated with HSA to activate carboxylic acids and react with amine groups. The solution was vortexed and placed on a rotator at room temperature to react for 4 h. The reaction was quenched with 20 mg of glycine for 20 min. The HSA-conjugated particle solution was then centrifuged at 15,000 rpm for 5 min. The supernatant was aspirated, and the pellet was reacted with avidin or stained to measure the amount of HSA.

The amount of HSA on the particle surface was measured using an anti-HSA antibody conjugated with APC fluorophore (R&D Systems, Minneapolis, MN, USA). A mouse IgG2A antibody conjugated with APC was used as the isotype control. First, HSA-coated particles were washed with flow buffer (FB, PBS+/+ with 1% bovine serum albumin (BSA), pH 7.4) to remove unreacted HSA followed by centrifugation at 15,000 rpm for 5 min. Next, 1 × 10^6^ HSA-coated particles were resuspended in 0.1 mL of FB with 10 μL of antibody stain. The particles were stained for 15 min and washed with FB. The median fluorescence of stained HSA-coated particles was measured on an Attune flow cytometer (Thermo Fisher Scientific, Waltham, MA, USA). A set of APC calibration beads (Bang Laboratories, Fishers, IN, USA) were run on a flow cytometer the same day to create a curve correlating median fluorescence to the number of fluorescent molecules present, as previously reported [32]. The adjusted median fluorescence of particles (fluorescence after staining with anti-HSA-APC antibody minus fluorescence after staining with isotype APC antibody) was converted to a total number of HSA sites using the calibration curve slope and normalized to particle surface area.

### 2.3. Avidin Conjugation and Targeting Ligand Attachment

All PLGA particle types were conjugated with avidin utilizing carbodiimide chemistry as previously described [15]. Briefly, avidin (NeutrAvidin, ThermoFisher, Waltham, MA, USA) was dissolved in filtered deionized water. Particles were dispersed in filtered deionized water and added to the avidin solution to a volume of 0.5 mL. The particle/avidin solution was rotated for 20 min. Next, EDAC was dissolved in 200 mM MES at a pH of 4.5 for uncoated PLGA or pH 7.2 for coated PLGA. A total of 0.5 mL of EDAC/MES solution was added to the particle/avidin mixture. The final solution was rotated and reacted for 4 h. The reaction was quenched with 20 mg of glycine as above. Avidin-conjugated particles were collected by centrifugation and used the following day. The amount of avidin was measured with biotin-FITC using flow cytometry.

Particle coating with adhesion ligand was achieved by incubating avidin-conjugated particles for 30 min with the desired concentration of biotinylated sialyl Lewis A (sLe^A^, Glycotech Corporation, Rockville, MD, USA) or human anti-ICAM-1 antibody (aICAM1, R&D Systems) in FB. The targeted particles were then washed with FB and stained with respective antibodies for site density characterization. The sLe^A^ surface density on particles was quantified using an anti-CLA-FITC (Miltenyi Biotec, Bergisch Gladbach, Germany) antibody for 15 min, with Rat IgM FITC used as an isotype control. For aICAM1 density determination, goat anti-mouse IgG FITC (Jackson ImmunoResearch Laboratories, West Grove, PA, USA) and goat IgG FITC (isotype) were used. Control and stained particles were run on a flow cytometer (Attune) to measure median fluorescence intensity. A set of FITC calibration beads (Bangs Laboratories) was used to create a calibration curve to correlate fluorescence intensity to the number of molecules on the particle surface. The slope obtained from the calibration curve converted adjusted median fluorescence intensities to the number of avidin sites or targeting moiety and was normalized to the surface area.

### 2.4. Particle Surface Characterization

The surface morphology of particles was characterized by scanning electron microscopy (SEM). For SEM, lyophilized particles were resuspended in deionized water and dried on glass slides. Before imaging, the glass slides were mounted on SEM stubs and sputter-coated with gold. SEM images were analyzed using ImageJ to determine mean particle size. The zeta potential was measured using the Malvern Zetasizer Nano ZS, where particles were tested in deionized water. X-ray photoelectron spectroscopy (XPS) was used to chemically characterize the surface of CSPLGA by mounting dried samples onto indium foil. The infrared spectra of dried PLGA, CSPLGA, HSA-PLGA, and HSA-CSPLGA were obtained using ATR-FTIR on a diamond crystal. An average of 32 scans were taken per sample at a resolution of 4 cm^−1^ from 600 to 4000 cm^−1^.

### 2.5. HUVEC Culture

Human umbilical vein endothelial cells (HUVEC) were harvested from umbilical cords donated by the Motts Children’s Hospital at the University of Michigan under a Medical School Internal Review Board (IRB-MED)-approved human tissue transfer protocol (HUM00026898). This protocol is exempt from informed consent per federal exemption category #4 of the 45 CFR 46.101(b). HUVEC were isolated using a collagenase perfusion method, pooled, and grown in T75 flasks pretreated with 0.2% gelatin at 37 °C and 5% CO_2_ until confluent, as previously described [33]. The HUVEC flasks were trypsinized and seeded onto 30 mm glass coverslips pretreated with gelatin crosslinked with glutaraldehyde. The seeded HUVEC were incubated for approximately 48 h at 37 °C and 5% CO_2_ until confluent. HUVEC monolayers were activated with 2 mL of 1 ng/mL interleukin-1β (IL-1β) for 4 and 24 h to induce high E-selectin and intercellular cell adhesion molecule-1 (ICAM-1) expression, respectively.

### 2.6. Blood Preparation

Venous blood was obtained from healthy donors with written consent following a protocol approved by the University of Michigan Internal Review Board. Anticoagulant, either acid-citrate dextrose (ACD) or heparin (HEP), was added to blood at 0.14 anticoagulant to whole blood ratio. Next, blood was centrifuged at 2250× *g* for 7 min to separate the blood into red blood cells (RBCs), white blood cells (WBCs), and plasma. The plasma was placed into a clean centrifuge tube, the WBCs were removed, and RBCs were washed with PBS−/−. The plasma and RBCs were centrifuged at 2250× *g* for 7 min to remove any remaining WBCs. Anticoagulant-free plasma (ACF) was used in some assays to create a more physiologically relevant condition. For these assays, whole blood with no anticoagulant was centrifuged after being drawn as described above and quickly isolated following the same centrifugation steps for anticoagulated plasma. ACF plasma was used immediately in flow adhesion assays or SDS-PAGE protein characterization before visual clotting. All samples were kept at 37 °C before usage, except for ACF plasma.

### 2.7. Parallel Plate Flow Chamber Setup

Flow adhesion experiments were conducted using a parallel plate flow chamber (PPFC) (Glycotech Corporation) assay with a rectangular channel gasket [17]. For each experiment, a confluent monolayer of HUVEC activated for 4 or 24 h, as previously described, was vacuum-sealed to the bottom of PPFC. After conjugation with targeting ligand, particles were resuspended in FB, and concentration was measured using a hemocytometer. Targeted particles were added at 5 × 10^5^ particles per mL to a mixture of RBCs in FB or plasma (ACD, HEP, or ACF) at a fixed hematocrit of 40%. The particle mixture was perfused through the chamber for 5 min of laminar flow at a shear rate of 200, 500, or 1000 s^−1^ controlled by a syringe pump. The particle samples in ACD or HEP plasma conditions were preincubated for 5 min before being added to the chamber. To measure base particle binding, particles were added to a mixture of RBCs in FB and perfused into the chamber without incubation. For ACF plasma experiments, particles were mixed with ACF plasma/RBCs and immediately added to the chamber at the same flow conditions before any significant visual clotting. After each experiment, FB was perfused through the chamber, and 10 fluorescent images of the HUVEC monolayer along the chamber’s center were captured. Bound particles were counted using ImageJ. Control adhesion experiments were conducted with untargeted particles (unconjugated and avidin conjugated particles) exposed to inactivated HUVEC in similar flow conditions, as described above.

### 2.8. SDS-PAGE Gel Preparation

The adsorption of plasma proteins onto the particle surface was qualitatively characterized using SDS-PAGE. First, sLe^A^-coated PLGA, HSA-PLGA, CSPLGA, and HSA-CSPLGA particles were prepared, as described above. Next, 2.5 × 10^6^ sLe^A^ targeted particles were resuspended in 150 µL of PBS−/− and sonicated at 20% amplitude to disperse. A 500 µL volume of 100% plasma (ACD, HEP, or ACF) was added to the particle solution and incubated for 5 min at 37 °C to mimic flow experiment conditions detailed above. Following incubation, particles were washed with PBS−/− to remove unbound proteins and resuspended in 50 µL of 1X lane marker non-reducing buffer (Thermo Scientific Pierce, Waltham, MA, USA). The particle solution was heated to 95 °C for 5 min using a thermocycler to denature and release surface-bound proteins. The resulting solution was centrifuged at 15,000 rpm for 5 min. Next, 25 µL of supernatant from each sample was pipetted into individual wells on 4–20% Tris-Glycine protein gels (Invitrogen Novex WedgeWell, Waltham, MA, USA). The gel ran for approximately 30 min at 200 V along with a standard molecular weight ladder (Precision Dual Color Protein Standard, Bio-Rad Laboratories, Hercules, CA, USA) for comparison. The gel is removed from the cast and stained with Coomassie blue (Invitrogen SimplyBlue SafeStain) overnight. Gels were washed in deionized water to de-stain and image.

### 2.9. Statistical Methods

The data from flow adhesion experiments are plotted as adhesion efficiency, the ratio of particles bound in RBC-in-plasma over RBC-in-FB. For protein corona characterization, SDS-PAGE gels were analyzed with ImageJ. Each n represents an individual donor. The data on all figures and tables are plotted with standard error bars. The statistical analysis was conducted using Prism for each dataset and specified in the captions. Both adhesion efficiencies and band intensities were analyzed using one-way ANOVA with Dunnett’s multiple comparisons test, with PLGA as control. Statistical significance is displayed as * = *p* < 0.05, ** = *p* < 0.01, *** = *p* < 0.001, **** = *p* < 0.0001, and ns = not significant.

## 3. Results

### 3.1. Characterization of HSA and CS-Coated PLGA

PLGA and CSPLGA microparticles fabricated using the emulsion solvent evaporation method had HSA coupled to their surface via covalent chemistry. The SEM images of both unloaded and rhodamine-loaded particles fabricated are shown in Appendix A, and particle size and zeta potential are displayed in Appendix A. The resulting diameters for PLGA and CSPLGA microparticles were all approximately 1.6 µm. All microparticles had a spherical shape and a smooth surface, as shown in Appendix A. Here, unloaded PLGA had a negative charge of about −30.3 mV, while unloaded CSPLGA demonstrated a shift to approximately +10.7 mV. The shift in zeta potential from net negative to positive suggests CS successfully coated PLGA particle surface, given the known cationic nature of CS [34]. The size of PLGA and CSPLGA did not change significantly with rhodamine encapsulation. However, there were slight shifts (+/−3–4 mV) in the zeta potential for the rhodamine-loaded particles, to −26.2 mV and +7.18 mV for PLGA and CSPLGA, respectively (Appendix A).

The full spectra and the nitrogen N1s spectrum regions for both unloaded and rhodamine-loaded PLGA and CSPLGA obtained from XPS are shown in Appendix A. The nitrogen peak was apparent for unloaded CSPLGA, confirming the presence of CS on the PLGA surface. Rhodamine-loaded CSPLGA did not show a prominent nitrogen peak. Additionally, the infrared spectra using ATR-FTIR were obtained as a secondary method for detecting CS on PLGA and are displayed in Appendix A. The typical peaks between 1500 and 1600 cm^−1^ corresponding to amide I and amide II on CS were absent for unloaded and rhodamine-loaded CSPLGA. There was a faint peak at the 3500 cm^−1^ for unloaded and rhodamine-loaded CSPLGA particles corresponding to N-H stretch (see Appendix A). Although it was challenging to detect CS with XPS and ATR-FTIR for rhodamine-loaded CSPLGA, the shift in surface charge from −26.2 mV to +7.18 mV (Appendix A) and similarities of the obtained IR peaks to ones in a prior publication with PLGA/CS composite fibers suggests the presence of CS on the particle surface [35].

After fabrication, HSA was conjugated to the surface of microparticles. The HSA measured was approximately 21,000 and 122,000 sites/µm^2^ on PLGA and CSPLGA, respectively (Table 1). Next, particles were conjugated with the protein avidin, allowing the attachment of any biotinylated ligands or antibodies for targeting cell adhesion molecules expressed on the vascular wall. After the avidin reaction, the number of HSA sites detected was approximately 14,000 and 44,000 sites/µm^2^ on PLGA and CSPLGA (Table 1), respectively, as determined by flow cytometry. The ATR-FITR spectra of HSA-PLGA and HSA-CSPLGA clearly showed the presence of amide I and amide II peaks, demonstrating the successful conjugation of HSA onto the particle surface (see black arrows in Appendix A). A schematic of the final targeted particles to clearly show various coating schemes is depicted in Figure 1a.

### 3.2. Adhesion of HSA- and CS-Coated PLGA Particles Targeted with sLe^A^

The binding of PLGA, HSA-PLGA, CSPLGA, and HSA-CSPLGA to an inflamed endothelial cell monolayer was evaluated using a parallel plate flow chamber (PPFC) assay. Particles were conjugated with biotinylated sialyl Lewis A (sLe^A^), a ligand that binds to E-selectin expressed by endothelial cells during inflammation. All particles were conjugated with 1000 sites/µm^2^ of sLe^A^, as reported in Table 1. The binding of sLe^A^-coated particles in RBC-in-flow buffer (FB) and RBC-in-plasma was compared since earlier work showed loss of PLGA particle adhesion in the presence of plasma proteins. Representative images of sLe^A^-targeted particles bound to inflamed endothelial cells at a shear rate of 200 s^−1^ are shown in Figure 1b. The quantified raw particle adhesion density for all sLe^A^ particles is displayed in Appendix A. All particle types had similar adhesion levels in the RBC-in-FB condition at this low shear rate.

For assays with particles in RBC-in-plasma, there was a significant reduction in adhesion density for sLe^A^-coated PLGA and CSPLGA relative to binding in plasma-free conditions (RBC-in-FB) as shown in Figure 1b. To quantify the results, the number of particles bound in RBC-in-plasma was normalized to their binding in RBC-in-FB, defined here as adhesion efficiency and plotted in Figure 1c–e. At 200 s^−1^, the adhesion efficiencies of sLe^A^-coated PLGA and CSPLGA particles were 23% and 13%, respectively. Both showed a drastic reduction in binding, suggesting that adding CS onto PLGA alone does not mitigate the previously reported negative impact of plasma proteins on PLGA particle adhesion. Conversely, both sLe^A^-coated HSA-PLGA and HSA-CSPLGA particles retained a significantly higher level of binding in plasma. HSA-PLGA particles experienced double the adhesion efficiency (46%) compared to PLGA (*p* = 0.0459). HSA-CSPLGA particles saw the highest amount of binding in plasma with 78% adhesion efficiency, a threefold increase compared to PLGA (*p* < 0.0001).

Particle binding was also evaluated at 500 and 1000 s^−1^ to assess whether increasing the shear rate would affect particle adhesion efficiency. The adhesion efficiencies of all sLe^A^ particle types did not change significantly with the increase in the wall shear rate from 200 to 500 s^−1^. When the shear rate increased to 1000 s^−1^, all sLe^A^-coated particles, except for HSA-CSPLGA, showed no significant change in adhesion efficiencies. For sLe^A^-coated HSA-CSPLGA, the adhesion efficiency dropped to 46%, significantly different from its binding at 200 s^−1^ (*p* = 0.0003). Despite this reduced adhesion efficiency relative to their low shear adhesion, sLe^A^-coated HSA-CSPLGA still observed a fourfold higher adhesion efficiency at 1000 s^−1^ than PLGA with *p* = 0.0001. Overall, the HSA-CS coating on PLGA recovered the most targeted particle adhesion in plasma relative to bare PLGA across all shear rates evaluated.

Lastly, the adhesion of untargeted particles to an inactivated endothelium was measured to learn whether the particle binding recovery of HSA-PLGA and HSA-CSPLGA is due to specific ligand-receptor interactions. As shown in Appendix A, the particle adhesion density of untargeted particles in RBC-in-plasma at 200 s^−1^ was minimal for all avidin conjugated particles, except for HSA-CSPLGA. HSA-CSPLGA with no avidin showed similar binding to avidin-conjugated HSA-CSPLGA in RBC-in-plasma, suggesting some nonspecific interactions at low shear due to the addition of HSA. When the shear rate increased to 1000 s^−1^, the adhesion of avidin conjugated HSA-CSPLGA was slightly reduced, whereas all other untargeted particles had almost no binding in RBC-in-plasma.

### 3.3. Alternative Ligand Schemes on HSA-CSPLGA to Improve Binding at High Shear

sLe^A^-targeted has-CSPLGA particles demonstrated the most improvement in adhesion efficiency over PLGA at low and intermediate shear rates but were reduced at 1000 s^−1^. Thus, alternative targeting schemes were explored to determine whether receptor-ligand kinetics can impact the adhesion efficiency at high shear. During inflammation, endothelial cells also overexpress intercellular cell adhesion molecule 1 (ICAM-1), which is involved in the firm attachment of leukocytes to the endothelium. Here, a biotinylated anti-ICAM1 (aICAM) antibody was conjugated to HSA-CSPLGA particles at a density of 5700 sites/µm^2^ on average. The quantified adhesion of aICAM-targeted particles at 1000 s^−1^ is shown in Figure 2a. The adhesion efficiencies of aICAM-targeted PLGA, CSPLGA, and HSA-CSPLGA were 37%, 23%, and 41%, respectively. There was no significant difference for aICAM HSA-CSPLGA over bare PLGA (*p* = 0.8015). Next, the amount of aICAM sites was increased to about 10,000 sites/µm^2^ to understand whether that would promote firm adhesion of HSA-CSPLGA when exposed to plasma. The increase in aICAM sites shifted adhesion efficiency to 52% but was not statistically significant (*p* = 0.1381) compared to 5300 sites of aICAM on PLGA (Figure 2b). Given that dual targeting is another approach to improve particle binding affinity by using two ligands in synergy to target multiple receptors, HSA-CSPLGA was conjugated with 4000 sites/µm^2^ of sLe^A^ plus 6300 sites/µm^2^ of aICAM to target both E-selectin and ICAM-1 on the activated endothelium. The adhesion efficiency of sLe^A^ + aICAM-targeted HSA-CSPLGA particles was 72%, representing a significant increase over aICAM only PLGA (*p* = 0.0007). Overall, only the dual-targeted HSA-CSPLGA served to improve the adhesion efficiency of the HSA-CSPLGA in the high shear flow.

### 3.4. Characterization of Plasma Protein Corona on Coated PLGA with sLe^A^

Prior work suggests that the negative particle binding experienced by PLGA is linked to the protein corona acquired onto the particle surface. Thus, the protein corona on all particle types was characterized using SDS-PAGE to examine whether changes in the adsorption of plasma proteins are driving differences in particle binding. sLe^A^-targeted particles were incubated in ACD plasma for 5 min at 37 °C to mimic flow experiments. Images of representative gels of sLe^A^-targeted particles incubated in FB and ACD plasma are shown in Figure 3. Visually, there is a decrease in protein adsorption on sLe^A^ PLGA particles with CS surface coating, as indicated by fainter or absent protein bands. The sLe^A^ PLGA particles coated with HSA or HSA-CS showed increased protein adsorption, i.e., bolder band intensities across various molecular weights. However, some areas of interest showed distinct differences between particle types. sLe^A^-targeted HSA-PLGA and HSA-CSPLGA showed a significant increase at the 10–25, 50–75, and 76–150 kDa molecular weight ranges compared to PLGA. Given that the 10–25 kDa molecular weight band also appears in gels of HSA-PLGA and HSA-CSPLGA exposed to buffer, the proteins in this region most likely did not influence the differences in particle binding (Figure 3a).

The lanes between 50–75 and 76–150 kDa molecular weight ranges were analyzed using ImageJ, relating band intensity to the peak area. These results are plotted in Figure 3c–e. The first band above 50 kDa is the albumin band. Figure 3c directly compares the albumin band intensity, showing a slight increase for all particles compared to PLGA but that were not statistically different. Next, the band intensity at about 75 kDa for HSA-PLGA and HSA-CSPLGA showed a twofold (*p* = 0.0127) and threefold (*p* = 0.0010) increase compared to PLGA, respectively (Figure 3d). The band at approximately 150 kDa could correspond to IgG (≈150 kDa) or IgA (≈160 kDa), immunoglobulins of similar molecular weights. Figure 3e compares the band intensity at 150 kDa for all particle types. HSA-PLGA showed more than a threefold increase in intensity than PLGA (*p* = 0.0288), while HSA-CSPLGA had a fourfold increase (*p* = 0.0038). The increases in band intensity at the 75 kDa and 150 kDa molecular weight ranges potentially influenced the improved adhesion of HSA-conjugated particles in plasma.

### 3.5. Impact of Anticoagulant on Binding and Protein Adsorption of Coated PLGA

The choice of anticoagulants can impact particle binding and protein adsorption since they inhibit clotting through different mechanisms. Up to this point, ACD was used in the experiments detailed above, which works by chelating calcium. Heparin is another anticoagulant that binds to antithrombin to prevent clotting, affecting protein interactions. Thus, the adhesion of sLe^A^-coated HSA, CS, or HSA-CS-PLGA was evaluated in flow assays with plasma obtained with heparin as the anticoagulant. sLe^A^-targeted particles were exposed to RBC-in-HEP plasma for 5 min and perfused over an activated EC monolayer at 500 s^−1^. As shown in Figure 4, all sLe^A^-targeted particles except for HSA-CSPLGA had minimal binding in heparinized plasma. The adhesion efficiency of sLe^A^ HSA-CSPLGA is 10 times higher than PLGA (*p* = 0.0017). Since there are some differences in the binding of particles exposed to plasma with heparin as the anticoagulant, SDS-PAGE analysis was conducted as before and shown in Figure 4b. There was a slight increase in intensity overall for particles coated with HSA. The areas of interest are shown in Figure 4c,d. There was about a 1.6-fold (*p* = 0.4213) and 2.3-fold (*p* = 0.0604) increase in band intensity for HSA-PLGA and HSA-CSPLGA over PLGA, respectively, but they were not statistically significant. When the band at 150 kDa was isolated, there was a threefold increase in band intensity for HSA-PLGA compared to PLGA in heparin plasma (*p* = 0.0002) and an eightfold increase for HSA-CSPLGA (*p* < 0.0001).

Lastly, sLe^A^-coated particles were incubated in ACF plasma to model an environment closer to physiological conditions. The adhesion efficiency of sLe^A^ particles exposed to RBC-in-ACF plasma is shown in Appendix A. All particles had a significant reduction in binding in ACF plasma compared to ACD plasma. The adhesion efficiency of sLe^A^ HSA-CSPLGA was more than six times higher than PLGA, demonstrating some improvement (*p* = 0.0017) in these conditions. SDS-PAGE was conducted for particles exposed to ACF plasma for 5 min at 37 °C (Appendix A). The differences in protein bands were less clear when sLe^A^-targeted particles were exposed to ACF plasma than that of anticoagulated plasma. There was no significant difference at the 75 and 150 kDa bands as plotted in Appendix A. HSA-CSPLGA showed an around twofold increase at the 75 kDa band compared to PLGA (*p* = 0.3382). Interestingly, the adhesion efficiency of HSA-CSPLGA decreased when the anticoagulant changed from ACD to ACF. The differences in band intensity also decreased at 75 and 150 kDa in those plasma conditions.

## 4. Discussion

A limited number of polymeric particle systems have been successfully translated into the market for clinical use due to a lack of understanding of biological interactions [9,36]. For example, when drug carriers enter the bloodstream, the rapid adsorption of plasma proteins onto their surface impacts their biological fate, such as clearance rate, biodistribution, and targeting efficacy [37,38,39,40]. Physicochemical properties of drug carriers can affect the formation and composition of the protein corona, such as surface charge and hydrophilicity [41]. Indeed, previous studies demonstrated that plasma protein adsorption on PLGA particles drastically reduced their vascular wall adhesion in human blood—a necessary feature for vascular-targeted drug carriers designed to leverage cellular disease markers on the endothelium for disease site recognition [17,18,19,20]. Given their continued appeal for use as drug carriers, the high adsorption of unfavorable proteins (i.e., immunoglobulins) onto PLGA particles is a critical issue that needs to be addressed before its successful clinical translation. While it is known that surface modification of drug carriers can influence the amount and types of proteins that adsorb onto the surface, leading to favorable biological interactions, such as reduced cellular uptake or increased circulation time for various particle types [42,43,44], PEGylation—the gold standard for minimizing protein adsorption—did not improve the vascular targeting efficacy of PLGA to an inflamed endothelial [17,18,19,20]. Thus, there is a need for alternative surface coatings for polymeric carriers such as PLGA. In this work, we explored the use of chitosan (CS) or albumin (typically from human serum, HSA) coatings, independently or in combination, given that several studies have shown that coating PLGA with CS or HSA has reduced protein adsorption and increased circulation times [45,46,47]. We hypothesized that CS, HSA, or HSA-CS on PLGA particle surfaces could reduce or alter protein adsorption, improving binding to an inflamed endothelium for vascular targeting.

PLGA microparticles were successfully coated with CS and HSA via physical adsorption and covalent attachment, respectively. Fabrication conditions were adjusted to obtain spheres of about a 2 µm diameter, which have shown optimal targeting to blood vessel wall over nanosized carriers [15]. Typically, smaller microparticles below <6 µm can also avoid embolism, indicating the potential for clinical translation [48,49]. A shift in the PLGA particle surface charge, i.e., zeta potential, towards a positive value compared to plain PLGA particles’ strong negative charge suggests the successful adsorption of CS on the PLGA particle surface [50]. Additionally, XPS and ATR-FTIR spectra were employed to detect CS’s nitrogen atoms and N-H bonds. The presence of CS was clear for unloaded CSPLGA, but there was a decrease in the N1s peak intensity in the XPS spectra of rhodamine-loaded CSPLGA. Here, the addition of the rhodamine could be affecting the CS solution properties (i.e., ionic strength) and decreasing adsorption onto PLGA, making it difficult to detect [51]. Additionally, the IR spectra of unloaded and rhodamine-loaded CS were identical, displaying a slight peak at 3500 cm^−1^ corresponding to the N-H stretch. However, the amide peak was not detectable, which could indicate that the amount of CS adsorbed onto the surface of PLGA was not enough to be observed from the spectra. Ultimately, the PLGA particles exposed to CS coating increased the amount of HSA covalently attached over uncoated PLGA, most likely due to the primary amines present at every CS monomer, allowing increased amine bond formation with carboxylic groups on HSA. Conversely, PLGA only has two carboxylic acid groups at each polymer end, thus limiting the number of HSA molecules that can be conjugated, despite many amine sites being available on the protein.

Interestingly, like uncoated PLGA, CSPLGA particles also showed a drastic reduction in binding in the presence of plasma proteins, suggesting that the addition of CS onto PLGA does not mitigate the previously reported negative impact of plasma proteins on vascular wall adhesion. Conversely, HSA-coated particles targeted with sLe^A^ experienced a significant improvement in binding to an inflamed endothelium, even in the presence of plasma proteins. sLe^A^-coated HSA-PLGA maintained the same ratio of particles bound in plasma to flow buffer across all shear rates tested. The particle type with the highest percentage of particles bound in plasma relative to flow buffer was sLe^A^-coated HSA-CSPLGA. At the low and intermediate shear rates, sLe^A^ HSA-CSPLGA outperformed HSA-PLGA by keeping 80% of its particle adhesion, even after exposure to plasma proteins, but was reduced to about 50% at high shear. However, the improvement of the vascular adhesion of PLGA particles via the dual HSA-CS coating was optimal in medium-to-low shear conditions, with only moderate increases in HSA-CSPLGA particle observed at high shear (1000 s^−1^), tested across two different targeting ligand schemes; we exchanged sLe^A^ for anti-ICAM1 to target ICAM-1 on the endothelium, which is involved in the firm arrest of leukocytes during inflammation. In general, the presented result demonstrated that the addition of HSA can enhance the adhesion of PLGA and CSPLGA. While the superior performance of the HSA-CS dual coating may suggest synergy, it is also likely that the amount of HSA plays a role since HSA-CSPLGA had more HSA sites compared to HSA-PLGA. An HSA site density-dependent enhanced binding of HSA-PLGA particles is plausible due to the known albumin interactions with albumin-specific receptors, such as gp60, on the endothelium [52].

An alternative approach to improving drug carriers’ binding is employing two ligands in synergy to enhance targeting by mimicking leukocytes’ rolling and firm adhesion. One study showed increased binding and uptake by endothelial cells when liposomes were decorated with antibodies for E-selectin and ICAM [53]. HSA-CSPLGA particles with a 2:3 ratio of sLe^A^ to aICAM experienced a 70% adhesion efficiency at high shear. The dual targeting approach resulted in the most significant increase in particle binding in plasma conditions at high shear, as sLe^A^ and aICAM1 coatings interact with multiple endothelial cell receptors—selectins and ICAM-1 expressed on endothelial cells [54]. This result is in line with a prior work showing the binding of dual (sLe^X^ and aICAM1)-targeted polystyrene was enhanced by the initial rolling initiated by sLe^X^ interactions with P-selectin [55], suggesting that having multiple interactions with the endothelium is beneficial for supporting the firm adhesion of particles.

The PLGA particle protein corona was characterized via SDS PAGE, given that changes in the protein corona of modified PLGA could be driving the differences in particle adhesion. Indeed, several other studies have focused on manipulating the protein corona by pre-coating particles with favorable proteins, such as HSA [56,57]. Two protein bands stood out in this analysis. There was a significant increase in band intensity at the 75 and 150 kDa molecular weight (MW) marks for particles coated with HSA. HSA-CSPLGA had the highest intensity at these two bands, likely influencing its enhanced binding in plasma. The first band of interest potentially consists of histidine-rich glycoprotein (HRG), with a molecular weight of about 75 kDa. HRG is present at relatively high concentrations in human plasma and has been shown to interact with several ligands [58]. Previously, HRG has been shown to act as a dysopsonin when it adsorbs onto the surface of silica nanoparticles, leading to a reduction in uptake by macrophages [59]. The presence of dysopsonin proteins can aid particulate carriers in evading phagocytes. The second band of interest is most likely composed of immunoglobulins (Ig), specifically IgA or IgG, with molecular weights around 150 kDa. This increase in the immunoglobulin band for the best binding particles is of particular interest since earlier work showed that the high adsorption of large molecular weight proteins reduces the binding of PLGA particles. However, when individual Igs were depleted from plasma, IgA and IgM were the main culprits in reducing particle binding [18]. Importantly, the depletion and re-addition of IgG to plasma did not significantly reduce particle binding, which suggests that IgG may be present on the surface of our HSA-CSPLGA particles.

Indeed, HRG has also been shown to interact with IgG in the formation and clearance of immune complexes [58]. It is possible that the increased presence of IgG on the HSA-CSPLGA is promoting the subsequent adsorption of HRG. Interestingly, when heparin was used as the anticoagulant, the adhesion of sLe^A^-targeted HSA-CSPLGA was slightly reduced compared to its binding in ACD plasma but was still significantly better than bare PLGA. This change in particle binding could be due to HRG interacting with heparin instead of the particle surface containing IgG, supported by a decrease in the 75 kDa band intensity in the particle corona derived for heparinized plasma [58]. Mass spectrometry and depletion assays could further confirm the identity of these proteins since they seem to play a role in improving particle adhesion. Lastly, HRG has been shown to interact with the CLEC-1A, C-type lectin domain family 1 member A, a receptor on endothelial cells [60]. The increase in HRG in the protein corona of HSA-coated particles creates another opportunity to interact with CLEC-1A on the endothelium.

## 5. Conclusions

In our study, we evaluated the effect of surface modification on PLGA particle adhesion in the presence of plasma proteins. Specifically, we focused on coating PLGA with CS, HSA, or both to improve biological interactions. Our results indicate that the addition of HSA to the surface of particles enhances the binding of PLGA to an inflamed endothelium even in the presence of plasma proteins. The amount of HSA on the particle surface could play a role in the level of particle adhesion since HSA-CSPLGA outperformed HSA-PLGA. Interestingly, the addition of CS alone onto PLGA showed no improvement, although there was a reduction in protein adsorption. The dual coating of CS and HSA onto PLGA experienced the highest level of improved particle binding. The improvement in adhesion efficiency depended on ligand type and targeting schemes, especially at higher shear. Increasing the number of interactions between drug carriers and the endothelium by conjugating multiple ligands or ligands with an affinity for multiple receptors could be necessary for enhanced particle binding. We infer that HRG and IgG, on the basis of molecular weight, have a higher presence on the surface of HSA-conjugated particles relative to bare PLGA, which could benefit particle adhesion. Ultimately, our work suggests that the adhesion of PLGA particles is possible when the protein corona is altered after surface modification with albumin, leading to favorable protein adsorption.

## Figures and Tables

**Figure 1 pharmaceutics-14-01018-f001:**
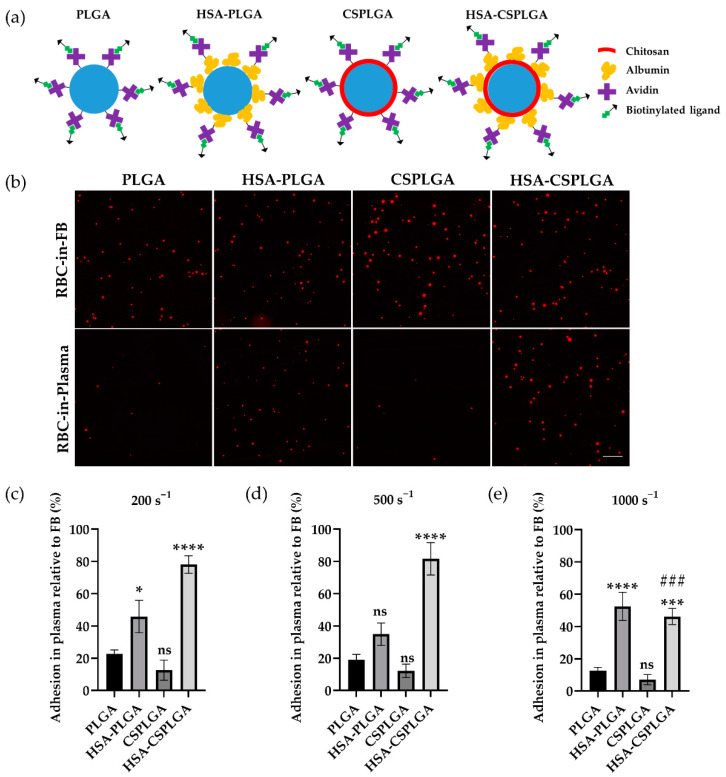
Adhesion of sLe^A^-targeted PLGA, HSA-PLGA, CSPLGA, and HSA-CSPLGA to an inflamed endothelial cell monolayer in vitro. (**a**) Schematic of ligand-targeted uncoated and coated PLGA. (**b**) Representative fluorescent images of rhodamine-loaded, sLe^A^-targeted particles bound to inflamed endothelial cells in RBC-in-FB and RBC-in-plasma conditions at 200 s^−1^. Particles with 1000 sites/µm^2^ of sLe^A^ were added to RBC-in-FB or RBC-in-ACD plasma at a 5 × 10^5^ particles/mL concentration and perfused over HUVEC activated for 4 h. Particles in plasma were incubated for 5 min prior to the experiment. Quantified adhesion of particles in RBC in plasma relative to RBC FB at 200, 500, and 1000 s^−1^ are shown in (**c**–**e**). Statistical analysis was completed using one-way ANOVA with Dunnett’s multiple comparison test with PLGA as control. * = *p* < 0.05, *** = *p* < 0.001, **** = *p* < 0.0001, and ns = not significant. ### = *p* < 0.001 relative to binding at 200 s^−1^. *n* = 10 distinct donors. Error bars represent standard error. The scale bar is 100 µm. sLeA = sialyl Lewis A, HSA = human serum albumin, CS = chitosan, ACD = acid-citrate-dextrose, RBC = red blood cell, FB = flow buffer.

**Figure 2 pharmaceutics-14-01018-f002:**
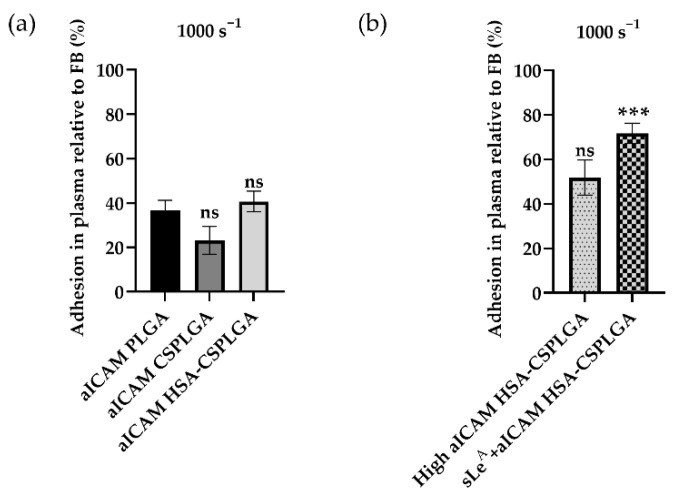
Alternative targeting schemes to improve binding of HSA-CSPLGA in ACD plasma at a high shear rate. (**a**) Binding of aICAM-1-targeted particles over endothelial cells activated for 24 h. Particles with ≈5700 sites aICAM-1/µm^2^ were added in RBC FB and RBC ACD plasma at a 5 × 10^5^ particles/mL concentration and perfused over HUVEC activated for 24 h. Particles in plasma were incubated for 5 min before the experiment. (**b**) Binding of particles with higher aICAM-1 site density (≈10,000) and dual-targeted HSA-CSPLGA (≈6000 aICAM-1 plus ≈5000 sLe^A^) after 24 h of HUVEC activation. Statistical analysis was completed using one-way ANOVA with Dunnett’s multiple comparison test with aICAM PLGA as control for A. *** = *p* < 0.001 and ns = not significant. *n* ≥ 9 distinct donors for A and B. Error bars represent standard error.

**Figure 3 pharmaceutics-14-01018-f003:**
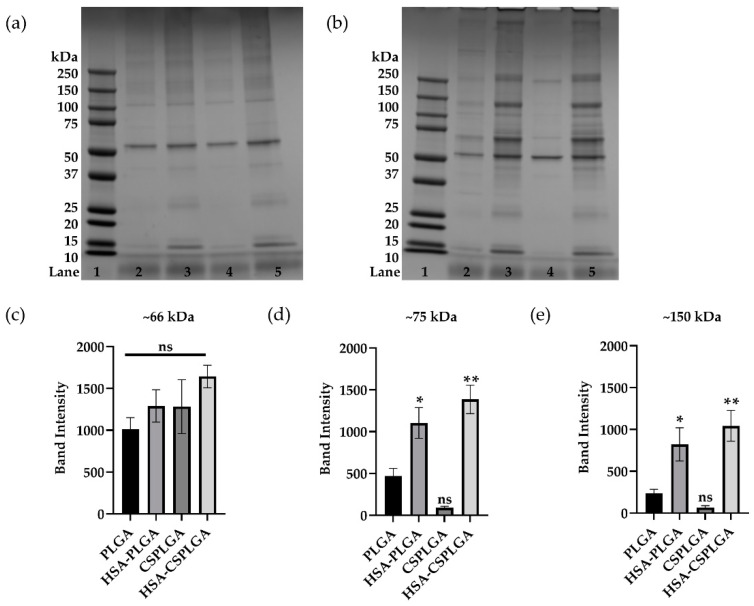
Protein corona characterization of sLe^A^-coated PLGA, HSA-PLGA, CSPLGA, and HSA-CSPLGA in flow buffer (FB) and ACD plasma. The 2.5 × 10^6^ particles conjugated with 1000 sites/µm^2^ of sLe^A^ were incubated in 650 µL of 78% ACD plasma for 5 min at 37C. SDS-PAGE of sLe^A^-targeted particles in FB (**a**) and ACD plasma (**b**). Lane 1: molecular weight ladder, Lane 2: PLGA, Lane 3: HSA-PLGA, Lane 4: CSPLGA, and Lane 5: HSA-CSPLGA. Each lane was analyzed with ImageJ. Plotted are isolated band intensities at ≈66 kDa (**c**), ≈75 kDa (**d**), and ≈150 kDa (**e**). Statistical analysis was completed using one-way ANOVA with Dunnett’s multiple comparison test with PLGA as control: * = *p* < 0.05, ** = *p* < 0.01, and ns = not significant. *n* = 4 distinct donors. Error bars represent standard error.

**Figure 4 pharmaceutics-14-01018-f004:**
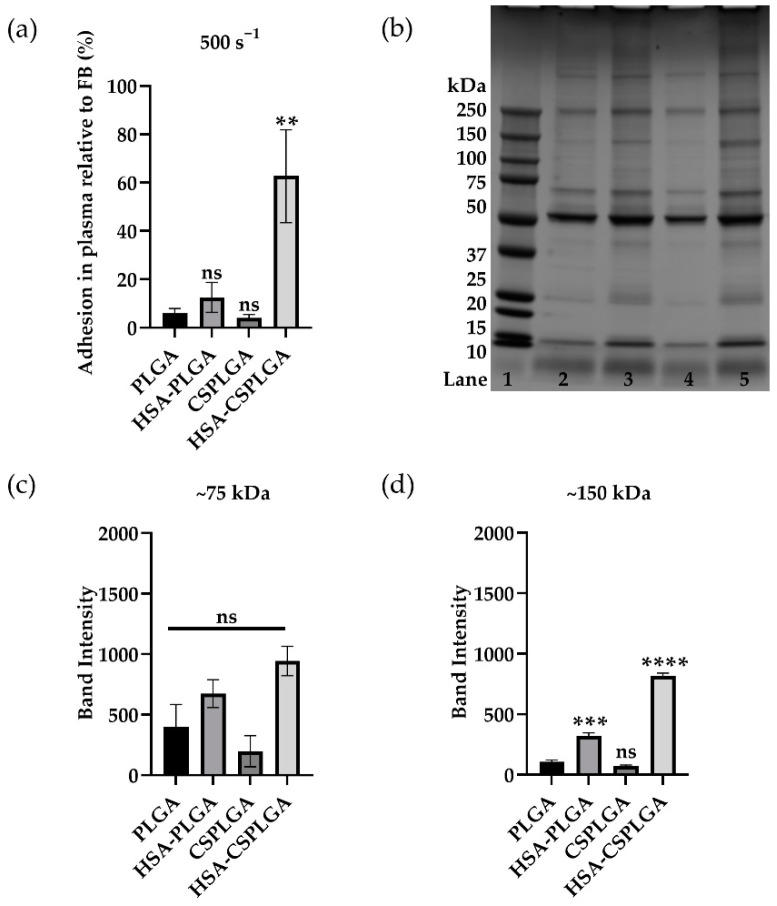
Adhesion and protein characterization of sLe^A^-coated PLGA, HSA-PLGA, CSPLGA, and HSA-CSPLGA in heparin plasma at 500 s^−1^. Particles with 1000 sites/µm^2^ of sLe^A^ were incubated in RBC FB and RBC plasma at a 5 × 10^5^ particles/mL concentration and perfused over HUVEC activated for 4 h. Particles in plasma were incubated for 5 min before the experiment. (**a**) Particle adhesion efficiency in heparin plasma. (**b**) SDS PAGE of particles exposed to heparin plasma. Lane 1: molecular weight ladder, Lane 2: PLGA, Lane 3: HSA-PLGA, Lane 4: CSPLGA, and Lane 5: HSA-CSPLGA. Each lane was analyzed with ImageJ. Plotted are isolated band intensities at ≈75 kDa (**c**) and ≈150 kDa (**d**). Statistical analysis was completed using one-way ANOVA with Dunnett’s multiple comparison test with PLGA as control: ** = *p* < 0.01, *** = *p* < 0.001, **** = *p* < 0.0001, and ns = not significant. *n* = 7 for (**a**) and *n* = 3 for (**c**,**d**) distinct donors. Error bars represent standard error.

**Table 1 pharmaceutics-14-01018-t001:** Protein and ligand surface site density was measured via flow cytometry.

Particle Surface Density (Sites/µm^2^)				
Particle Type	HSA [after Avidin]	Avidin	sLe^A^	aICAM
PLGA (sLe^A^ or aICAM)	N/A	8000 ± 3000	1200 ± 300	5300 ± 1300
HSA-PLGA (sLe^A^)	21,000 ± 8000 [14,000 ± 10,000]	14,000 ± 12,000	1000 ± 300	N/A
CSPLGA (sLe^A^ or aICAM)	N/A	18,000 ± 8000	1200 ± 300	5800 ± 1100
HSA-CSPLGA (sLe^A^ or aICAM)	122,000 ± 32,000 [44,000 ± 11,000]	32,000 ± 11,000	1300 ± 300	6000 ± 1300
HSA-CSPLGA (High aICAM)			N/A	10,000 ± 1500
HSA-CSPLGA (sLe^A^ + aICAM)			4000 ± 1200	6300 ± 1400

HSA = human serum albumin, CS = chitosan, sLe^A^ = sialyl Lewis A, aICAM = human anti-ICAM.

## Data Availability

Data are contained within the article.

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
