# Peer review of "Dual Coating of Chitosan and Albumin Negates the Protein Corona-Induced Reduced Vascular Adhesion of Targeted PLGA Microparticles in Human Blood"

_pharmaceutics, 2022, doi:10.3390/pharmaceutics14051018_

Round 1

Reviewer 1 Report

Dear Editor, in the submitted paper PLGA microparticles were coated with chitosan (CS), human serum albumin (HSA), or both (HSA-CS) to improve adhesion and to be used as vascular-targeted carriers (VTCs). The paper is well organized and contains new and interesting data and for this reason I propose to accept it for publication. In following you can find some minor importance comments.

A recent review in which the application of PLA and its copolymers for drug encapsulation in microparticles was recently published but it mishes form this work (please see Pharmaceutics 14, 359, 2022. https://doi.org/10.3390/pharmaceutics14020359).

The characteristics of used polymers and other materials are not given in the experimental part

Author Response

We thank the reviewer for their comment.  

  1. A recent review in which the application of PLA and its copolymers for drug encapsulation in microparticles was recently published but it mishes form this work (please see Pharmaceutics 14, 359, 2022. https://doi.org/10.3390/pharmaceutics14020359)

We are delighted to incorporate the recent review that the reviewer drew our attention to, as the article is indeed a relevant resource for future readers of our work.  The review article is included as reference #12 and cited as follows:

    • PLGA microparticles were fabricated using the emulsion solvent evaporation method established in the literature [12,30]. 
    • (PLGA) – a negatively charged polymer that is easily degraded via hydrolysis into its monomeric forms that are subsequently metabolized by the human body [10] – is the most widely studied material because of its tunable capability to encapsulate various drugs, ranging from small molecules to proteins [11,12]
  1. The characteristics of used polymers and other materials are not given in the experimental part.

We have included detailed characteristics of all materials used in the methods and materials section as customary for most journals where this section appears before the result section. We are happy to repeat this information throughout the results section if the Editors deem this necessary.

Reviewer 2 Report

I think it can be published as it is.

Author Response

We thank the reviewer for their thorough review of our work.

Reviewer 3 Report

The manuscript entitled “Dual coating of chitosan and albumin negates the protein…” by Lopez-Cazares describes the fabrication of chitosan and human serum albumin coated PLGA microparticles as vascular targeted carriers. This study examined the plasma protein-microparticle interactions as a strategy to target microparticles to the inflamed endothelium. The study was well-planned and thorough. Maybe, the authors could have tested the HSA-CSPLGA VTC in a small animal model to conclusively demonstrate the in vivo targeting efficacy.

  1. Figure 1. Adhesion of sLeA targeted PLGA, HSA-PLGA, CSPLGA, and HSA-CSPLGA to inflamed endothelium. Please revise "endothelium" to _endothelial cells in vitro._

Author Response

We thank the reviewer for their comment.  As requested, we have edited the title for Figure 1 to better reflect that the work was done with endothelial cells in culture.  

Change to manuscript

Figure 1. Adhesion of sLeA targeted PLGA, HSA-PLGA, CSPLGA, and HSA-CSPLGA to inflamed endothelial cell monolayer in vitro

Reviewer 4 Report

The manuscript entitled "Dual coating of chitosan and albumin negates the protein corona-induced reduced vascular adhesion of targeted PLGA microparticles in human blood" addresses the development and characterization of PLGA microparticles were coated with chitosan (CS), human serum albumin (HSA), or both (HSA-CS) to improve adhesion to inflamed endothelium in the presence of human plasma proteins. The work is well written, and I consider it valid for publication. Accept in present form.

Author Response

We thank the reviewer for their comment!

Reviewer 5 Report

Dear EiC,

Dear Authors,

I checked the manuscript carefully and, in my opinion the manuscript is very promising but, the core@shell (the coated nature) is not proved (just supposed). Even if you are using a specific method of synthesis and other researchers obtained coated structures, it is not sure you will obtain the same (even minor changes can be sufficient). So, I would like to ask you to prove the core@shell nature.

Best regards,

Anton 

Author Response

Reviewer Comment

I checked the manuscript carefully and, in my opinion the manuscript is very promising but, the core@shell (the coated nature) is not proved (just supposed). Even if you are using a specific method of synthesis and other researchers obtained coated structures, it is not sure you will obtain the same (even minor changes can be sufficient). So, I would like to ask you to prove the core@shell nature.

Author Response

We thank the reviewer for their thorough review of this manuscript. The central goal of the work presented in the paper was to modify PLGA particles such that their surface targeting functionality is preserved in the presence of human plasma proteins.  We explored two possibilities - (1) PLGA particles with Chitosan used in the water phase for fabrication and (2) modification of the particle surface with albumin covalently bound. 

In response to our first round of reviews, we demonstrated the presence of chitosan on the particle surfaces according to the following data.

  1. The shift in particle zeta potential towards the positive direction, e.g., from -26.2 mV to +7.18 mV (Table S1), suggests the presence of chitosan (CS) on the particle surface since this polymer is highly positively charged, and it was the only polymer in the water phase during fabrication. PLGA micro-/nano-spheres are traditionally prepared via emulsion solvent extraction/evaporation (ESE) techniques using polyvinyl alcohol (PVA) as the choice of surfactant for stabilizing the polymer droplets in oil-in-water and water-in-oil-in-water emulsions. However, residual amounts of the surfactant are found in all particle formulations via the ESE method, even after several washing steps and subsequent lyophilization.  Here we used chitosan instead of PVA; hence we expect chitosan on the surface of particles caused the shift in zeta potential. [Sahoo, S.K., J. Panyam, S. Prabha, and V. Labhasetwar, Residual polyvinyl alcohol associated with poly (D, L-lactide-co-glycolide) nanoparticles affects their physical properties and cellular uptake. J Control Release, 2002. 82(1): p. 105-14.]
  2. We added details and characterization data to confirm the presence of our coatings, specifically FTIR spectra. In the literature, the main techniques used are XPS, FTIR, and NMR. Here, we obtained the spectra of chitosan-coated PLGA using XPS and ATR-FTIR. The measurement of surface charge is another method used to detect the presence of chitosan due to its cationic nature. It was challenging to detect chitosan after rhodamine encapsulation using XPS and ATR-FTIR, potentially due to not having enough coating. We did see a similar shift in surface charge for chitosan-coated PLGA with and without rhodamine with a 3-4 mV difference, which demonstrates chitosan's presence.  
  3. We now reference (#35) a published article that used a similar approach to detect chitosan coating on PLGA fiber and presented spectra with similar IR peaks to the ones we report in our manuscript. 

    Vaezifar, S.; Razavi, S.; Golozar, M.A.; Esfahani, H.Z.; Morshed, M.; Karbasi, S. Characterization of PLGA/Chitosan Electrospun Nano-Biocomposite Fabricated by Two Different Methods. http://dx.doi.org/10.1080/00914037.2014.886244 2014, 64, 64–75, doi:10.1080/00914037.2014.886244.

    We hope these additional clarifications would be sufficient to demonstrate that the chitosan used in the water phase for particle fabrication was incorporated onto the PLGA particle surface, as typical for any polymer present in the water phase for the solvent evaporation method. Notably, the presence of the chitosan polymer led to enhanced adhesive function of the particles.

Modification to manuscript

On page 7: Though it was challenging to detect CS with XPS and ATR-FTIR for rhodamine-loaded CSPLGA, the shift in surface charge from -26.2 mV to +7.18 mV (Table S1) and similarities of the obtained IR peaks to ones in a prior publication with PLGA/CS composite fibers suggests the presence of CS on the particle surface [35].